# Pericapsular Nerves Group (PENG) Block in Children under Five Years of Age for Analgesia in Surgery for Hip Dysplasia: Case Report

**DOI:** 10.3390/jpm13030454

**Published:** 2023-02-28

**Authors:** Małgorzata Domagalska, Katarzyna Wieczorowska-Tobis, Tomasz Reysner, Alicja Geisler-Wojciechowska, Monika Grochowicka, Grzegorz Kowalski

**Affiliations:** 1Chair and Department of Palliative Medicine, University of Medical Sciences, 61-701 Poznań, Poland; 2Department of Anesthesiology and Intensive Care, W.Dega Orthopedic and Rehabilitation Clinical Hospital, University of Medical Sciences, 61-545 Poznań, Poland

**Keywords:** pain, myelomeningocele, paralytic dislocation of the hip, congenital dislocation of the hip, Dega transiliac osteotomy

## Abstract

Introduction: The Pericapsular Nerve Group (PENG) block is a novel technique that allows for analgesia of the anterior hip capsule via the articular branches of the accessory obturator nerve and femoral nerve, which have a significant role in the innervation of the hip capsule. A PENG (Pericapsular Nerves Group) blockade is effective in both adult and pediatric patients. However, no studies on patients under five are available in the literature. Herein, we describe our experience with two pediatric patients with hip dysplasia. Purpose: This study aimed to evaluate the analgesic effect of the pericapsular nerves group (PENG) in preschool children undergoing hip surgery. Patients and methods: This study included two patients, aged 4 and 2 years old, who were qualified for hip surgery. Spinal or general anesthesia with the addition of a PENG block was performed. During the procedure, the basic hemodynamic parameters were monitored. The pain was assessed using the FALCC (Face, Legs, Activity, Cry, Consolability scale) score. A dose of 15 mg/kg^−1^ of metamizole was administered if the FLACC score was 3. In the case of a score of 4 on the FLACC scale, the application of 0.2 mg/kg^−1^ of nalbuphine was ordered. Results: After the surgery, the patients received 15 mg/kg^−1^ IV paracetamol every 6 h to prevent rebound pain. The patient’s hemodynamic parameters were stable and within normal range. In the first 24 h period, the FLACC scores from all patients ranged from 0 to 3. One patient required metamizole 12 h after surgery. No evidence of block complications was observed. Conclusions: This case series showed that the PENG block assured opioid-free pain management and provided adequate postoperative analgesia. However, we are convinced that future randomized, controlled trials are needed in this field.

## 1. Introduction

The use of local anesthesia for perioperative analgesia in hip surgery is a widespread practice that has been shown to reduce pain, perioperative complications, and postoperative opioid use [1]. In pediatric anesthesia, regional techniques are the mainstay of perioperative analgesia, either alone or as part of a multimodal analgesia strategy. Perioperative pain management is paramount after pediatric hip surgery. Inadequate analgesia can lead to patient and parent dissatisfaction, longer recovery, and prolonged hospital stays [2]. Local anesthesia is beneficial because it relieves pain and reduces opioid side effects [3,4].

Neural axis technology reduces pain scores in pediatric patients undergoing hip replacement surgery. However, positioning requirements, bilateral sensory and motor blocks, and urinary retention limit their use. Caudal and lumbar plexus blocks are the most common regional blocs, which are advanced techniques. However, many anesthesiologists are reluctant to use a lumbar plexus block because of the risk of complications such as hematoma, renal puncture, and high neuraxial anesthesia [5]. Conversely, the caudal block is often not optimally suited due to limited efficacy and duration of action. Therefore, our institutional practice includes the administration of spinal anesthesia with propofol sedation or general anesthesia with propofol and remifentanil, both regimens combined with an epidural catheter for children undergoing unilateral or bilateral hip surgery. This type of anesthesia effectively blocks nerves that innervate the hip. However, sometimes the analgesia is insufficient due to epidural catheter dislodgement or migration [6]. The use of an epidural catheter is also limited due to the need for continuous hemodynamic monitoring during postoperative analgesia [7].

Girón-Arango et al. [8] first described the Pericapsular Nerve Group (PENG) block in 2018 for perioperative pain management in patients with hip fractures. This block was confirmed by a cadaveric dye study [9] that exhibited the pericapsular expanse aiming only at the sensory arms of the anterior hip capsule with a motor-sparing effect. PENG block stops innervation of the anterior hip capsule broadly. In addition, this advanced ultrasound-guided technology numbs the obturator, paraobturator, and sensory capsular branches of the femoral nerve [9,10].

Unfortunately, there are no studies on patients under five years of age in the available literature. Therefore, we present using the PENG block for perioperative pain management in two pediatric patients undergoing unilateral surgery for pediatric hip dysplasia.

## 2. Material and Methods

Written informed consent was obtained from the parents for this scientific contribution.

Patients were admitted to W. Dega Orthopedic and Rehabilitation Clinical Hospital at Poznan University of Medical Sciences with a hip pathology as a part of a multidisciplinary care pathway. 

After discussion with the surgical team, treatment goals included 2 days of hospitalization in the PICU (Pediatric Intensive Care Unit) for postoperative pain relief and the complete removal of motor blockade. In addition, we suggested a PENG block for postoperative pain relief instead of an epidural catheter. In both cases, after the induction of anesthesia, the PENG block was performed (Figure 1 and Figure 2). 

The patients were kept supine during the focal block. A low-frequency linear transducer (2–5 MHz) was put in a transverse plane along the left anterior inferior iliac spine (AIIS) to identify the iliopsoas muscle, femoral nerve, and femoral artery (FA). The probe was then rotated counterclockwise to align with the pubic ramus to visualize the AIIS, iliopsoas prominence (IPE), FA, iliopsoas, and iliopsoas notch. Under direct vision, a 22-gauge 50 mm echo needle was advanced in the lateral-medial plane between the psoas tendon and the pubic ramus until the needle tip touched the IPE. The needle was withdrawn, and after negative aspiration, 0.2% ropivacaine was injected in incremental doses. Preparation and surgical positioning allowed for maximal time for block onset. 

The basic hemodynamic parameters were monitored postoperatively during the procedure and the first 24 h. After the surgery, the patients received 15 mg/kg^−1^ IV paracetamol every 6 h to prevent rebound pain. The pain was measured every 2 h using a FLACC score. A dose of 15 mg/kg^−1^ of metamizole was administered if the FLACC score was 3. In the case of a FLACC score ≥ 4, an application of 0.2 mg/kg^−1^ of nalbuphine was ordered. The evidence of block complications, like mobility disorders, bleeding, neuropathy, or systemic toxicity from local anesthetic deposition, was observed. Muscle weakness or mobility of the hip joint was difficult to follow due to the plaster cast.


**1st Case Report**


A 4-year-old, 14.3 kg, boy with developmental dysplasia due to myelomeningocele and paralytic dislocation of the left hip joint, with no other comorbidities (Table 1), was walking independently and did not require urine catheterization. Apart from the primary disease, he was not hospitalized. The boy did not take medications permanently. He had no drug allergies and no history of convulsions. Due to his primary disease, he was under the care of an orthopedic, rehabilitation, and urological clinic. In June 2020, he underwent surgery for meningomyelocele, and in October 2020, he had an Achilles tenotomy on the left side. The course of the procedures and anesthesia occurred without complications. His neurological development was expected according to age. He qualified for Dega transiliac osteotomy with femoral subtrochanteric osteotomy combined with a Mustard iliopsoas transfer. The ASA physical status classification was 3. An hour before the planned surgery, 7.5 mg of midazolam po and 4 mg of IV dexamethasone were administered. In addition, 200 mg of paracetamol was given intravenously before surgery. General anesthesia was performed with a continuous infusion of propofol and remifentanil in doses, ensuring BIS was within the limits of 40–60 and airway potency was maintained using a laryngeal mask. The lung-sparing mechanical ventilation with Air/O_2_ mixture was used, under the control of SpO_2_ > 94% and pCO_2_, within the 35–45 mmHg range (Getinge, FlowC, Maquet Critical Care AB, Sweden). After the induction of general anesthesia, the PENG block was performed with 7.5 mL of 0.2% Ropivacaine (Table 1). During the procedure, the hemodynamic parameters (MX550, Philips Medizin Systeme Beoblingen GmbH Hewlett-Packard-Str. 2, Germany) and the remifentanil dosage were monitored.

In the first 24 h postoperative, the patient’s hemodynamic parameters were stable and within the normal range according to age. After the surgery, the pain was measured every 2 h using a FLACC score. A 15 mg/kg^−1^ metamizole was administered if the FLACC score was 3. In the case of a FLACC score ≥ 4, the application of 0.2 mg/kg^−1^ of Nalpain was ordered. There was no evidence of block complications, like mobility disorders, bleeding, neuropathy, or systemic toxicity from local anesthetic deposition. Muscle weakness or mobility of the hip joint was difficult to observe due to the plaster cast. However, parents and caregivers did not keep any mobility or sensory disturbances.


**2nd Case Report**


A 22-month-old, 18kg girl with congenital hip joint dislocation with no other comorbidities was walking independently and was not hospitalized apart from the primary disease. The girl did not take medications permanently. She had no drug allergies and no history of convulsions. She was under orthopedic and rehabilitation care due to her primary disease. In August 2020, she underwent hip arthrography, closed hip repositioning, and plaster cast placement. In September 2020, chronic instability of the left hip joint and burns in the places of skin folds in the groin was found in the operating room conditions. The plaster cast was removed, and it was decided that surgery was necessary. After 7 days, an open reposition of the hip joint with K-wire stabilization was performed. In October 2020, the K-wire was removed, and in November, the plaster cast was cut through, and a Frejka pillow was adjusted. The course of the procedures and anesthesia occurred without complications. Her neurological development was expected according to age. The girl was qualified for transiliac osteotomy with femoral subtrochanteric osteotomy. The ASA physical status classification was 2 (Table 1). An hour before the planned surgery, 7.5 mg of midazolam po and 4 mg of IV dexamethasone were given.

In addition, 250 mg of paracetamol was given intravenously before surgery. Sedation was performed with continuous propofol infusion at 5mg/kg/hour. Spontaneous ventilation was maintained with an oxygen mask at 2 L/min. Spinal anesthesia (L3/4, PAJUNK, sprotte needle 27 G, 70 mm) was performed with 1.5 mL of 0.5% heavy spinal bupivacaine. After the spinal anesthesia, the PENG block was performed with 5 mL of 0.2% ropivacaine (Table 1).

In the first 24 h postoperative, the patient’s hemodynamic parameters (MX550, Philips Medizin Systeme Beoblingen GmbH Hewlett-Packard-Str. 2, Germany) were stable and within the normal range. After the surgery, the pain was measured every 2 h using a FLACC score. A dose of 15 mg/kg^−1^ of metamizole was administered if the FLACC score was 3. In the FLACC score ≥ 4, the application of 0.2 mg/kg^−1^ of nalbuphine was ordered. There was no evidence of block complications, like mobility disorders, bleeding, neuropathy, or systemic toxicity from local anesthetic deposition. Muscle weakness or mobility of the hip joint was difficult to observe due to the plaster cast. However, parents and caregivers did not observe any mobility or sensory disturbances.

## 3. Results

The patient’s hemodynamic parameters were stable and within normal range. During the surgery, neither patient required extra doses of opioids. The time of the procedure was 105 and 75 minutes; the difference was due to the surgical technique. Blood loss was 200 mL and 170 mL (Table 2). 

Postoperatively, the patients appeared comfortable with numeric pain scores of 2/3 of 10. Paracetamol was administered every 6 h to prevent rebound pain. Overnight, both patients did not require breakthrough opioids or muscle relaxants. After 12 h postoperative, 200 mg of IV metamizole was administered to the male patient due to a FLACC score of 3. Later, FLACC and subjective pain scores were 0–1/10 for the next 24 h, and the parents did not report poor pain control (Table 2). On the first postoperative day, the patients could participate in physical therapy. 

The patients were discharged on the first evening following surgery with an acetaminophen-opioid preparation for pain control and as-needed diazepam for muscle spasms. There was no proof of block complications, like mobility disorders, bleeding, neuropathy, or systemic toxicity from large-volume local anesthetic deposition [11].

## 4. Discussion

Effective management and control of intraoperative and postoperative pain are essential in perioperative hip disease to minimize opioid use and its side effects. In our facility, most pediatric hip surgeries are performed under spinal anesthesia and propofol sedation, with preserved spontaneous ventilation. However, the 4-year-old boy had general anesthesia due to the myelomeningocele [12,13]. In addition, spinal and epidural anesthesia is inappropriate for patients with spinal malformations [14].

One of the difficulties in controlling hip pain using adequate regional analgesia is the complex innervation of the joint by multiple nerves. The most common local anesthesia and analgesia techniques for hip surgery are the lumbar plexus block, lumbar epidurals, femoral nerve blocks, sciatic nerve blocks, fascia iliac blocks, and pericapsular injections. However, they may provide only partial analgesia or lower extremity weakness, hypotension, and related side effects, especially in patients with congenital and acquired musculoskeletal system defects [14].

The single-shot PENG block has lately been reported in the cadaveric study [15,16] and in the literature for perioperative pain management in hip surgery by aiming the articular arms of the accessory obturator nerve (AON), femoral nerve (FN), and obturator nerve (ON) [8,17,18,19].

The technical simplicity of imaging in traction-fixed patients and no need for multiple punctures made this blockage ideal for young pediatric patients [10]. Unfortunately, there are no studies regarding PENG block in patients younger than five years old, such as in our case report. 

The PENG block has become a widely used ultra-sound guided regional technique for facilitated motion-preserving hip blocks. However, quadriceps weakness was observed after the PENG block [20]. The precise mechanism of femoral nerve involvement after PENG blockade results from local anesthetic diffusion through the plane between the pectoralis major and the major psoas muscle or into the muscle [21]. The involvement of the femoral nerve in pediatric anesthesia for hip surgery may be desirable, especially in surgeries involving femoral osteotomy. Also, the motor-sparing effect is not essential due to the plaster cast after surgery. Çiftçi et al. [16], in their cadaveric study, showed that 30 mL of dye resulted in a more extensive spread around the femoral nerve trance from the inguinal to the knee, around the femoral cutaneous nerve and obturator nerve, compared to 20 mL. Based on the Çiftçi et al. cadaveric study, other pediatric case reports of older children [10,17,18,19,20,21,22], and calculating not exceeding the maximal dosage of ropivacaine, we decided to use two doses of ropivacaine, 0.3 mL/kg, and 0.5 mL/kg. 

Similar to our study, the pediatric case reports of older children showed the opioid-sparing effect of PENG block in the first 24 h postoperative [8,17,18]. In our research, parents and caregivers did not observe muscle weakness, as reported in other Polish studies [9,22,23], which is particularly important in children with myelomeningocele and other neuromuscular diseases neurodegenerative or other congenital nerve diseases [23].

A significant disadvantage of single-shot regional nerve blocks is the restricted time window of the analgesic effect. In addition, the potential for rebound hyperalgesia after a single nerve block from 12 to 24 h has been described [24,25]. In our study, children were given acetaminophen before surgery to avoid rebound pain.

Several additives, such as alpha-2-adrenergic receptor agonists and dexamethasone, have been shown to extend analgesia following single-dose regional nerve blocks [26]. Therefore, in our study, we decided to use dexamethasone with the best-proven effectiveness [27,28]. 

Also, some studies [29,30,31] have reported that few patients experience pain in the lateral femoral cutaneous nerve region after hip surgery. However, we did not observe it in our study.

In 2021, Morrison et al. [32] published a systematic review of studies describing the use of the PENG blockade in adults and pediatrics to treat hip pain caused by either fracture or surgery. They found 20 studies that met the inclusion criteria for both the PENG blockade alone and the PENG blockade in combination with other topical analgesic techniques. They concluded that the PENG blockade is a promising regional analgesic technique. We achieved similar results regarding analgesic efficacy and avoidance of rescue opioids.

## 5. Conclusions

Coherent with the literature, our experience established exercise-preserving and opioid-sparing pain management using PENG blocks in patients under five. Therefore, we settled on a safe and well-tolerated pediatric range of ropivacaine 0.3–0.5 mL of 0.2% ropivacaine [2]. We identify that the efficacy of PENG blocks has yet to be demonstrated in prospective clinical trials in the pediatric population. Nevertheless, our observations were encouraging, but future studies of PENG efficacy and safety in the pediatric population are warranted.

## Figures and Tables

**Figure 1 jpm-13-00454-f001:**
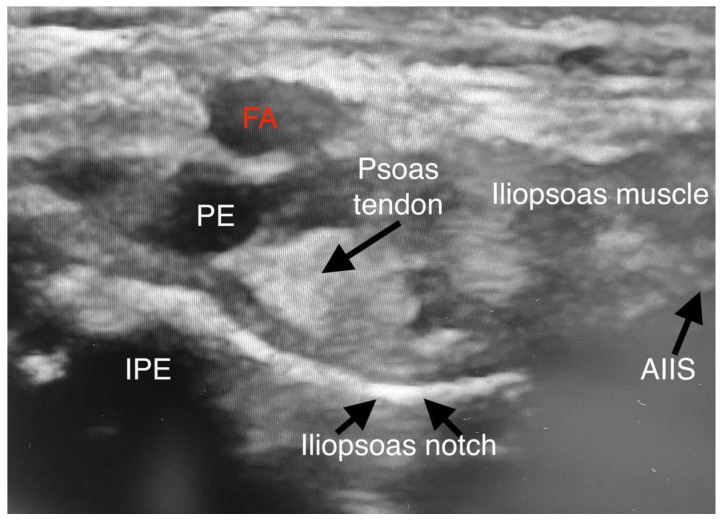
Sonoanatomy of PENG. (FA-femoral artery; PE—pectineus muscle; AIIS—anterior inferior iliac spine; IPE—iliopsoas prominence).

**Figure 2 jpm-13-00454-f002:**
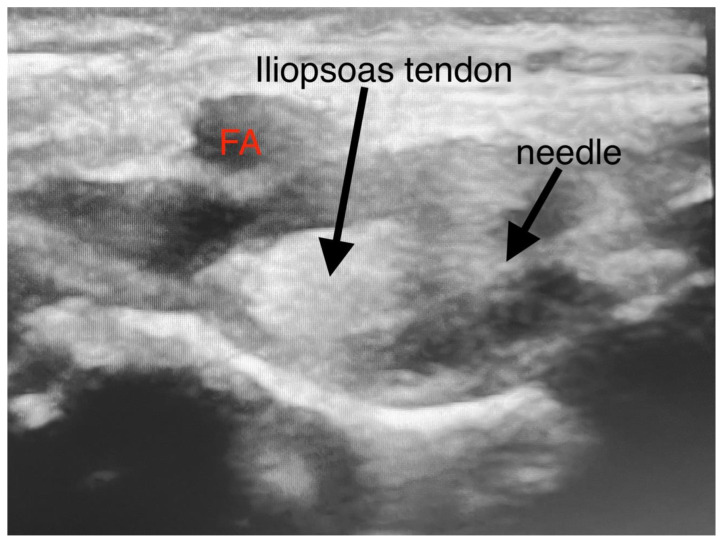
Injection technique of PENG block. (FA—femoral artery).

**Table 1 jpm-13-00454-t001:** Characteristics of patients.

	1st	2nd
Sex	M	F
Age	4 years	22 months
Weight	14.3 kg	18 kg
ASA	3	2
Comorbidities	Myelomeningocele	Obesity
Hip pathology	Paralytic dislocation of the left hip joint	Congenital dislocation of the left hip joint
Type of Surgery	Dega transiliac osteotomy with femoral subtrochanteric osteotomy combined with iliopsoas transfer, according to Mustard	Transiliac osteotomy with femoral subtrochanteric osteotomy
Type of Anesthesia	General anesthesia Propofol 0.2 mg/kg/min Remifentanil 0.1 ug/kg/min	Spinal anesthesia 1.5 mL heavy, spinal bupivacaine Propofol 0.06 mg/kg/min
Type and Volume of Local Anesthetic used for PENG block	7.5 mL of 0.2% ropivacaine	5 mL of 0.2% ropivacaine

**Table 2 jpm-13-00454-t002:** Surgery and postoperative course.

	1st	2nd
Total operating time	105 min	75 min
Complications	no	no
Estimated blood loss	200 mL	170 mL
Highest FLACC score	3	2
Pain medications	200 mg paracetamol every 6 h	250 mg paracetamol every 6 h
Interventional pain drugs	200 mg IV metamizole	no
Breakthrough opioids or muscle relaxants	no	no

## Data Availability

The data presented in this study are available on request from the corresponding author.

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
