# Peer review of "Pericapsular Nerves Group (PENG) Block in Children under Five Years of Age for Analgesia in Surgery for Hip Dysplasia: Case Report"

_jpm, 2023, doi:10.3390/jpm13030454_

Round 1

Reviewer 1 Report

It has been a pleasure to read the manuscript on pericapsular nerves group block. This case series presents two preschool children undergoing hip surgery with the use of PENG block. There were no adverse events and no need for rescue opioid analgesia revealing the PENG block as an effective and safe perioperative analgesia. The authors can be congratulated on their innovative approach to improving perioperative analgesia in children.

I would consider restructuring the methods section a potential for improvement of the manuscript. General information on PENG block and postoperative pain management including monitoring applies to both cases and should be mentioned as the first part of the methods section, followed by presentation of the details of both cases. In the current manuscript some information is mentioned redundantly, e. g. pain management P4 L90-97 and P4 L121-128.

In the following paragraphs I will make some further suggestions, which would improve the manuscript from my point of view.

Abstract

-          P1 L24: According to the main text, I assume that you meant opioid was administered when FLACC score was 4 or higher.

-          P1 L25: Please do not use trade names (nalbuphine instead of nalpain).

Introduction

-          P2 L46-48: I do not understand to which regional anaesthesia technique (caudal block or lumbar plexus block) refer the risk factors mentioned. Colonic and renal puncture are not considered typical risks of caudal block. I would rather say that caudal block is often not optimally suited due to the limited efficiency and duration of action, while the risk of complications is very low.

Material and Methods

-          P2 L70-72: The sentence seems not to be complete.

-          Without reading the main text the figures are difficult to understand, because abbreviations are not explained in the figure captions.

-          P5 L135: I do not know the term “tooth cast”, probably because I do not work in orthopaedic surgery. Please make sure, that the term is correctly translated.

-          P5 L134: Is the year correct (2022)?

-          P4 L 121 and P5 L152: This information would have been better put up in the results section.

Discussion

-          P6 L182: In my opinion this citation is not useful in this context. The paper cited covers procedural sedation and not surgical cases. Consider whether a citation is necessary at all.

-          P6 L185 and P7 L221: I think “topical” is not the correct term, but rather “local” or “regional”.

-          P7 L222: Please consider the wording in the last sentence of the discussion. There was no comparison of PENG block with other techniques or no regional anaesthesia. Therefore, it would be better to state “avoidance of rescue opioids”.

-          Conclusion: I would suggest discussing the dose of ropivacaine for PENG block in the main part of the discussion. How were the doses of ropivacaine administered calculated for both children? Why did the child with lower body weight receive more ropivacaine? What are doses used by other researchers performing PENG block?

Literature

-          There are some incomplete references, e. g. 10 and 11.

Language

-          The manuscript is professionally written, concise and easy to understand. Some language issues are present. So, the manuscript will benefit from language revision.

Author Response

The authors would like to thank the Reviewer for his comments. Care has been taken to improve the work and address his concerns as per the specific comments below.

P1 L24 and L25 "In the FLACC score 4, the application of 0,2mg/kg-1 Nalpain was ordered" has been changed to "In the case of 4 scores in the FLACC scale, the application of 0,2mg/kg-1 Nalpain was ordered."   P2 L46-46 "However, many anesthesiologists are reluctant to use this occlusion because of the risk of complications such as colonic and renal puncture and total obstruction" has been changed to " However, many anesthesiologists are reluctant to use lumbar plexus block because of the risk of complications such as hematoma, renal puncture, and high neuraxial anesthesia (5). Conversely, the caudal block is often not optimally suited due to limited efficacy and duration of action."   P2 L70-72 "Patients admitted to the W. Dega Orthopedic and Rehabilitation Clinical Hospital of Poznan University of Medical Sciences with a hip pathology as a part of a multidisciplinary care pathway. " has been changed to "Patients were admitted to the W. Dega Orthopedic and Rehabilitation Clinical Hospital of Poznan University of Medical Sciences with a hip pathology as a part of a multidisciplinary care pathway."   The abbreviations were added to figure captions.   P5 L134-135 "In August 2022, she underwent hip arthrography, closed repositioning of the hip, and placement of a tooth cast." has been changed to "In August 2020, she underwent hip arthrography, closed hip repositioning, and plaster cast placement."   P4 L121 and P5 L152 The information was moved to the results section.   P6 L182 We have deleted the citation.   P6 L185 and P7 L221 We have changed the name "topical" to "regional."   P2 L222 We have changed the sentence to "We achieved similar results regarding analgesic efficacy and avoidance of rescue opioids."   We have decided to add a description in the discussion section of why we have chosen to use two doses of ropivacaine.

Reviewer 2 Report

Good theme and certainly one of interest.

I have done a lot of PENG block in hip region and a cryneurolysis afterwards and have excellent results in doing that. While this is generally only case report I find it iinteresting because when it comes to postoperative analgesia in pediatric patients you often find that it sometimes lack in effectiveness.

Topic is excellent. The manuscript is well explained and easy to read for someone who does not do a lot of regional blocks.

Here is some hint from my experience - try using levobupivacaine because it has much more sensory blockade and much more better result and you use motor sparing block in the first place. I use ropivacaine also all the time but for ambulatory settings. For postoperative analgesia would try this.

Author Response

Thank You very much for the review.

We plan a randomized trial in pediatric patients undergoing hip surgery with PENG block as a part of a multimodal pain regimen. We will consider, as you suggested, using levobupivacaine, as you suggested, due to its more potent sensory blockade than ropivacaine.